# Spatial Assessment and Prediction of Urbanization in Maseru Using Earth Observation Data

Elhadi Adam [1],* , Nthabeleng E. Masupha [1] and Sifiso Xulu [2]

1   School of Geography, Archaeology and Environmental Studies, University of the Witwatersrand, Johannesburg 2025, South Africa
2   School of Agricultural, Earth and Environmental Sciences, University of KwaZulu-Natal, Westville Campus, Durban 4000, South Africa
*   Correspondence: elhadi.adam@wits.ac.za; Tel.: +27-117-176-532

**Abstract:** The availability of geospatial data infrastructure and earth observation technology can play an essential role in facilitating the monitoring of sustainable urban development. However, in most developing countries, a spatiotemporal evaluation of urban growth is still lacking. Maseru, Lesotho's capital and largest city, is growing rapidly due to various socioeconomic and demographic driving forces. However, urban expansion in developing countries has been characterized by entangled structures and trends exacerbating numerous negative consequences such as ecological degradation, the loss of green space, and pollution. Understanding the urban land use and land cover (LULC) dynamic is essential to mitigate such adverse impacts. This study focused on mapping and quantifying the urban extension in Maseru, using Landsat imagery from 1988 to 2019, based on the Support Vector Machines (SVM) classifier. We also simulated and predicted LULC changes for the year 2050 using the cellular automata model of an artificial neural network (ANN-CA). Our results showed a notable increase in the built-up area from 15.3% in 1988 to 48% in 2019 and bare soil from 12.3% to 35.3%, while decreased agricultural land (21.7 to 1.7%), grassland (43.3 to 10.5%) and forest vegetation (5.5 to 3.2%) were observed over the study period. The classified maps have high accuracy, between 88% and 95%. The ANN-CA projections for 2050 show that built-up areas will continue to increase with a decrease in agricultural fields, bare soil, grasslands, water bodies and woody vegetation. To our knowledge, this is the first detailed, long-term study to provide insights on urban growth to planners and other stakeholders in Maseru in order to improve the implementation of the Maseru 2050 urban plan.

**Keywords:** urbanization; land use and land cover (LULC) changes; ANN-CA; Landsat; change detection; prediction; Lesotho

## 1. Introduction

Urbanization has enormous impacts on changes in population characteristics and land use and land cover (LULC) class transformation [1]. More than half of the world's population lives in urban areas, and it is projected to grow by 2.5 billion between 2018 and 2050, with associated environmental repercussions [2]. In 2000, 26 percent of the world's population in low-income countries lived in urban areas, and this is expected to double by 2030 [3]. Most urban areas have experienced tremendous LULC changes due to rapid urbanization and urban growth caused by, among other things, an excess of births over deaths and internal and external migration [4]. Against this background, a thorough understanding of human-induced spatiotemporal LULC changes is required to manage environmental changes and improve urban sustainability [5].

Urbanization also plays an important role in the development of nations, as urban areas serve multiple functions in society and drive economic growth and technological advances [6]. Urbanization has led to the conversion of LULC features, such as forested areas and bare land, into built-up areas for residential and commercial use, roads, pavements,

and other modern infrastructure [7]. This transformation is often associated with the loss of agricultural land, pollution in the air, land and water, flooding, and the depletion of surface water bodies and groundwater sources [8]. Urbanization has negative effects on nations, serving as a breeding ground for unemployment, poverty, and inequality, among others [9]. It has also led to food insecurity [10], high crime rates, environmental degradation, the construction of unplanned settlements, and the uncontrolled haphazard growth of cities without proper planning, which has become a problem for governments and city dwellers [11]. However, it is also worth noting that with urbanization comes economic development and growth, especially when coupled with proper planning: a gain that is currently unrecognized in most countries of the Global South due to weak institutions and ill-resourced local planning authorities.

The effective management of urbanization resulting from LULC changes and associated environmental systems requires evidence-based approaches to mitigate and adopt undesirable changes [5]. The central city in Lesotho, Maseru, is situated just over the Caledon (Mohokare) River from South Africa. The city was first established as a police camp on the river's eastern edge following the 1869 Treaty of Aliwal North between the British Empire and the Orange Free State Boer Republic. Maseru has since grown as a small town to provide commercial, educational and health functions. After independence in 1966, the city underwent significant changes, including the growth of government buildings. Maseru eventually became home to 60% of Lesotho's urban population in 1986. The main factor of this rapid urbanization is a combination of natural increase and internal migration. The city recently introduced a 2050 urban plan that was established in 2017 for effective natural resources management through rehabilitating wetlands and rivers and restoring deteriorating landscapes to promote sustainable urban development per the sustainable development goals (SDGs) (https://www.gov.ls/maseru-2050-urban-plan/ accessed on 14 March 2022). Gathering evidence of urban LULC changes using traditional methods such as field surveys is time-consuming and resource-intensive [12]. This process requires the application of appropriate methods to analyze the drivers and impacts of urbanization on LULC over time. Remote sensing using both air- and space-borne sensors has provided an inexpensive, timeous, and effective way to analyze the impacts of urbanization on LULC changes over time and predict future urban growth [12,13]. This information is lacking in most metropolitan areas, particularly in sub–Saharan Africa. This limits the realization of sustainable cities and communities, preventing most urban areas in sub-Saharan Africa from achieving economic growth in terms of zero hunger, low poverty levels and job security. Therefore, it is crucial to understand the dynamics of these growth(s) in Sub-Saharan Africa and their trajectories to support proper planning and promote regional integration.

Several previous studies have analyzed the impact of urbanization on LULC and predicted urban growth using remote sensing techniques [14–17]. Wang and Maduako [15] analyzed urban growth for 31 years between 1984 and 2015 in Lagos, Nigeria. They predicted urban growth for 2050, finding massive changes in urban LULC and forecasts in built-up areas with a 25.14% (120,790 km$^2$) increase. Abudu et al. [14] quantified an urban sprawl between 2001 and 2016 and projected urban growth for 2021 in the Arua Municipality, Uganda. Their results showed an increase in built-up areas from 18.2% in 2001 to 40.9% of the total area.

There is growing interest in using the artificial neural network cellular automata (ANN-CA) to model spatiotemporal transitions and urban growth. This has been found to be effective for analyzing nonlinear complex LULC phenomena and avoiding the automatic acquisition of conversion rules during transitional computations [18,19]. Using self-organization, self-learning, association, and memory, ANN is capable of simplifying the acquisition of CA model conversion rules, extracting CA conversion rules from the original training data, and eliminating subjective factors, thus improving simulation accuracy [18]. Mansour et al. [16] analyzed spatiotemporal changes in the LULC between 2008 and 2018 and predicted urban expansion in 2038 using the MCA in the city of Nizwa, Oman. Their results showed that the city had changed by 418.5%, and by 2038 there would be an urban growth of 37,465 ha. Furthermore, the study projected a 10% decline in agricultural land and an increase of 6% in

built-up areas in 2031. Most of the previous research studies used various spatial models to estimate LULC changes with models such as the Land Transformation Model [18] and the CLUE (Conversion of Land Use and its Effects) model [19]. In the present study, we used ANN-CA in the city of Maseru, Lesotho, where no previous studies have been conducted to monitor the impact of urbanization on LULC changes and to predict future urban growth.

Urbanization has a positive as well as negative impact on LULC in countries in sub-Saharan Africa, which includes Lesotho [20]. Maseru is the capital of Lesotho, where a high rate of urbanization has led to massive urban growth, with the population doubling since the early 2000s [20], mainly due to high rates of immigration from other rural and urban places in Lesotho [21,22]. Maseru is Lesotho's most developed urban center, with high employment opportunities, public services, and infrastructure facilities [22]. One of the fundamentals of sustainable urban planning is the availability of LULC data to support the monitoring and forecasting of urban growth, which is crucial for decisions and policymaking [23]. Monitoring and forecasting city growth requires the use of historical and current remote sensing data for future city management [24]. A shortage of data often poses challenges in such regions. The lack of research studies in Maseru on the impact of urbanization on LULC changes and future urban growth brings a gap in the city's literature. In response, our objective in this research was: (i) to characterize LULC classes from 1988 to 2019 with Landsat data, (ii) to quantify LULC changes over the study period, and (iii) to simulate and forecast future urban growth using ANN-CA in the city of Maseru, Lesotho, for 2050.

## 2. Materials and Methods

### 2.1. Study Area

Maseru is the capital city of Lesotho: a mountainous country that is landlocked by South Africa and is considered the commercial hub of the country [25]. The city lies on the country's western border along the Caledon River, bordered by South Africa on the northwestern section (Figure 1). The city is mountainous, with topography ranging from 1482 m to 1905 m, as shown in Figure 1. Maseru has a continental and temperate climate receiving 85% annual rainfall between October and April, averaging 700 mm per year [26]. The city has lowlands and highlands, and temperatures include mild winters and hot summers [26]. The approximate average low temperature of the city is 8 °C, with a high average temperature of 22.4 °C [27].

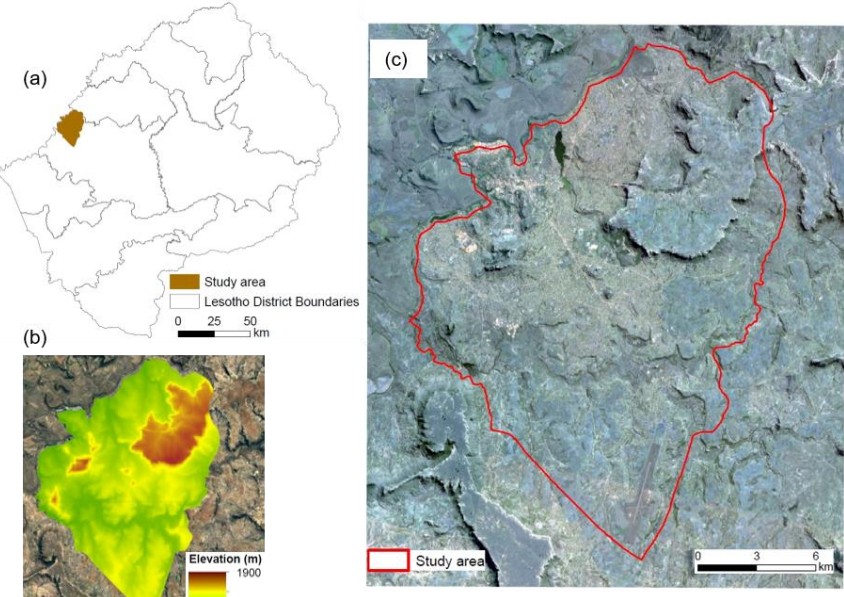

**Figure 1.** Map of (**a**) Lesotho, (**b**) A 30 m Shuttle Radar Topography Mission Digital Elevation Model (SRTM DEM) and (**c**) PlanetScope imagery of Maseru.

### 2.2. Remote Sensing Data Acquisition

Landsat imageries from 1988, 1993, 1998, 2003, 2008, 2013 and 2019 were used in this study to map and quantify LULC changes in the city of Maseru over three decades. These cloud-free imageries were downloaded from the United States Geological Survey (USGS) data repository (http://www.usgs.gov/ accessed on 15 October 2019). The images were acquired during the rainy season and between December and March to help distinguish between agricultural fields with bare soil and built-up areas (Table 1).

**Table 1.** Characteristics of Landsat imagery used in this study.

| Mission | Sensor | Date | Resolution | Path/Row |
|---------|--------|------|------------|----------|
| Landsat-5 | TM | 29 March 1988 | | |
| Landsat-5 | TM | 11 March 1993 | | |
| Landsat-5 | TM | 5 February 1998 | | |
| Landsat-5 | TM | 20 December 2003 | 30 m | 170/080 |
| Landsat-7 | TM | 20 March 2008 | | |
| Landsat-8 | OLI | 15 December 2013 | | |
| Landsat-8 | OLI | 14 January 2019 | | |

### 2.3. Image Pre-Processing

The multispectral bands for the Landsat imageries were calibrated to the correct imperfections encountered during image acquisition because of systemic errors, topographic errors, atmospheric conditions, and the earth-sun distance [28]. The radiometric calibration coefficients derived from the metadata files were used to convert the Landsat imageries to top-of-atmosphere (TOA). Then, an atmospheric correction was performed on Landsat imageries using the Fast Line-of-sight Atmospheric Analysis of Spectral Hypercubes (FLAASH) in ENVI 5.5 software. The geometric corrections were conducted on the Landsat imageries to avoid misalignment because of different viewing angles, which affect change detection accuracies [29]. The geometric correction based on the image-to-image registration was performed because the images were acquired from two different sensors. The Landsat imageries for 1988, 1993, 1998, 2003, 2008 and 2013 were registered to the 2019 imagery, producing a maximum error of 5% per tie point. All the corrected Landsat imageries were rectified to the Universal Transverse Mercator (UTM), World Geodetic System (WGS) 1984—Zone 35 South coordinate system.

### 2.4. Training and Test Samples

The LULC classes were defined from the Lesotho Land Cover Atlas developed by the Food and Agriculture Organization (FAO) of the United Nations [30]. The training and test points used to classify all the satellite imageries were collected from the agricultural field (AF), bare soil (BS), built-up (BU), grassland (GL), waterbody (WB) and woody vegetation (WV) classes. The points were collected with the help of aerial photographs, panchromatic images, and Google historical maps. To test the performance of the classification model, the sample points for the multitemporal images were divided into 70% (training) and 30% (test) for the classification of all the imageries (Table 2).

**Table 2.** Total number of training (TR) and test (TS) samples for all the classes (1988–2019).

| | | 1988 | | 1993 | | 1998 | | 2003 | | 2008 | | 2013 | | 2019 | |
|---|---|------|----|------|----|------|----|------|----|------|----|------|----|------|----|
| | | TR | TS | TR | TS | TR | TS | TR | TS | TR | TS | TR | TS | TR | TS |
| LULC classes | AF | 112 | 47 | 98 | 42 | 38 | 15 | 43 | 18 | 75 | 32 | 77 | 33 | 60 | 25 |
| | BS | 126 | 54 | 133 | 56 | 122 | 51 | 134 | 57 | 127 | 54 | 126 | 33 | 126 | 53 |
| | BU | 211 | 90 | 211 | 90 | 220 | 94 | 216 | 92 | 216 | 92 | 227 | 97 | 213 | 90 |
| | GL | 115 | 48 | 110 | 46 | 85 | 36 | 81 | 34 | 77 | 33 | 88 | 37 | 68 | 28 |
| | WB | 91 | 39 | 80 | 33 | 77 | 33 | 60 | 25 | 70 | 30 | 70 | 29 | 87 | 37 |
| | WV | 75 | 31 | 83 | 35 | 68 | 28 | 94 | 39 | 80 | 33 | 94 | 40 | 75 | 32 |

Agricultural field (AF), bare soil (BS), built-up (BU), grassland (GL), waterbody (WB), woody vegetation (WV).

### 2.5. Image Classification

The LULC classifications were computed using a supervised Support Vector Machine (SVM) classifier. The SVM is one of the most effective classifiers in remote sensing studies producing high accuracies in the classification of satellite imageries [31,32]. The SVM is a non-parametric machine learning classifier based on statistical models developed by Vapnik in the 1960s [33]. It seeks to define an optimum hyperplane to separate two classes using the training samples and then validate its generalization ability [34]. The classifier generates a decision boundary with large margins, with the smallest distance between the samples to the separation boundary [35]. It was also facilitated by kernel functions which could be applied to solve optimization problems, especially in high-dimensional spaces [34]. When using the SVM classifier, the training samples (70% of all classes in this study) were mapped onto a new feature space using a kernel function, and the SVM produced a large margin separation between the training samples for all the classes in a new feature space. With labeled sequences $(x_1, y_1), \ldots, (x_m, y_m)$ where $x$ stands for the covariates and $y \in \{-1, 1\}$ denotes the response, a kernel function ($k$) was used by SVM as illustrated in the following equation:

$$f(x) = \sum_{i=1}^{m} a_i k(x_i, x) + b, \tag{1}$$

where $a_i$ and $b$ coefficients are estimated through minimizing the function:

$$\sum_{i,j=1}^{m} a_i a_j k(x_i, x_j) + C \sum_{i=1}^{m} \varsigma_i, \tag{2}$$

which is subject to:

$$y_i f(x_i) \geq 1 - \varsigma_i, \tag{3}$$

where $C$ stands for the penalty parameter of misclassification and $\varsigma_i$ measures the degree of misclassification of $x_i$ [36]. The training vectors ($x$) are mapped onto a higher or infinite dimensional space using the function $f(x)$. The SVM classifier determines a linear hyperplane, separating the training samples a with maximized margin that is in the higher dimensional space [35]. In non-linear data, the dataset is transformed into a higher dimensional feature space using kernel-based functions such as radial basis, linear, sigmoid and polynomial [33]. In previous remote sensing studies, the radial basis was regarded as the best kernel function for classifying and mapping LULC features [37,38]. The radial basis requires tuning the cost and gamma parameters, which affects classification accuracies [32]. The best parameters were chosen using a 10-fold cross-validation method. The best cost and gamma parameters used in this study differed for years.

### 2.6. Accuracy Assessment

We used confusion matrices to validate classified maps for all the study years. The matrices were compared to the classified pixels of the test samples. The confusion matrices consist of the overall accuracy, kappa index, producer's, and user's accuracies. The overall accuracy is the proportion of samples that are correctly classified. In contrast, the producer's accuracy is the LULC map's accuracy from the view of the maker or the map producer [33]. The user's accuracy is defined as the viewpoint of the map user, and the kappa index assesses the agreement between the test data and the classifier [39]. The kappa index can be calculated using the following equation:

$$K = \frac{(f_o - f_E)}{(N - f_E)} \tag{4}$$

where $f_o$ denotes the number of observed agreements, $f_E$ stands for the number of agreements that are by chance, and the total number of observations is represented by $N$ [40]. If

the observations are in total agreement, then *K* is 1, and if there is no agreement among the observations other than the one expected by chance (denoted by $f_E$), then $K \leq 0$ [41].

### 2.7. Change Detection

In this study, we implemented the post-classification change detection exercise: a method mostly used to quantify LULC changes. It involves two images from different years of the same location, where changes are computed on a pixel-by-pixel basis to determine similarities and differences in pixel values [14]. We performed this exercise using the MOLUSCE plugin tool in the QGIS Desktop v.2.18.10 software. The MOLUSCE model conducts an analysis of LULC changes between the two time periods, examines the LULC transition potential and simulates future urban growth. The LULC change between two time periods can be calculated using the following equation:

$$\text{LULC change} = t_2 - t_1 \tag{5}$$

where $t_2$ is the final period, whilst $t_1$ stands for the initial period [42]. The changes were calculated between 1988 and 1993; 1993 and 1998; 1998 and 2003; 2003 and 2008; 2008 and 2013; 2013 and 2019 for all the classes used in this study.

### 2.8. Future Urban Growth Simulation

The multi-criteria evaluation (MCE), weights of evidence (WOE), artificial neural network (ANN), and logistic regression (LR) algorithm models were incorporated into the MOLUSCE plugin for potential transition modeling [43]. In this study, we applied the ANN for potential transition modeling, followed by the cellular automata (CA) for the LULC simulation. The ANN determines the transition probability of LULC using numerous output neurons in simulating LULC changes [44]. The training procedure in ANN is a process in which the connection weights are iteratively modified to accomplish a specific task [45]. This training process is called supervised because the desired outputs and all the input parameters are specified for each example [46]. The ANN represents a set of inputs that is a real-valued vector $[x_{1,...,}x_n]$. The error is the difference between the output results and the desired output desired by the ANN [46]. The back-propagation algorithm is applied after the training process to minimize the network error function by adjusting the weight values [47]. The error function is calculated using the following equation:

$$E = \frac{1}{2} \sum_{j-1}^{L} (d_j - o_j)^2 \tag{6}$$

where *L* represents the number of nodes in the later output and $d_j$ and $o_j$ represents the desired output and current response of node *j*, respectively [48]. In an iterative or repetitive approach, the corrections made to the weight parameters are computed and added to the previous values as follows:

$$\begin{cases} \Delta w_{i,j} = \eta \frac{\partial E}{\partial w_{i,j}} \\ \Delta w_{i,j}(t+1) = \Delta w_{i,j} + \alpha \Delta w_{i,j}(t) \end{cases} \tag{7}$$

where the weight parameter between nodes *i* and *j* is represented by $w_{i,j}$, $\eta$ represents the positive constant that controls the amount of adjustment, *t* is the number of iterations and $\alpha$ denotes the momentum factor, which takes on values in the range of 0 and 1. This parameter is also known as the stabilizing factor because it smoothes the rapid changes between weights [47].

We used the MOLUSCE model to simulate urban expansion in the study area for 31 years, from 2019 to 2050. The CA is a cell-based model that extrapolates historical and current LULC for the prediction of future changes using transition rules that are based on the status of the cells and their neighbors [41]. The state of each cell in the future is predicted

based on the historical state of the neighboring cells [16]. The performance of the model was validated to determine the correctness level. The classified or reference map was compared with the simulated map to determine the level of agreement using the validation table. A model of accuracy was presented using the kappa location, overall kappa, percentage correctness and kappa histogram. The ANN and CA approaches were used because they produced good results in the modeling and prediction of LULC changes [24,49,50].

### 2.9. Spatial Variables

Several factors control urban growth; these include spatial variables such as slope, elevation, water bodies, the existing urban land area, roads and population density, which influence the shaping pattern of future urban growth [50]. In this study, we used slopes, roads, existing built-up areas and water bodies as spatial variables to determine urban growth. Steep slopes are known to be less suitable for urban growth, and the lower the slopes, the more suitable they are for urban expansion [16]. Following Wang and Maduako [15] and Abudu et al. [14], we considered a slope below 25% as suitable for built-up developments and gave it a value of 1, while a value of 0 was given to a slope above 25% because they were unsuitable. We excluded areas very close to roads as no further developments were expected. For this reason, we created a 50 m buffer along and around the roads, and areas within the buffer were given a value of 0 (unsuitable), and areas located more than 50 m were given a value of 1, as they were suitable. We gave a value of 0 to existing built-up areas, as they were unsuitable for further growth and already developed, and a value of 1 was given to undeveloped areas. We gave a value of 0 to existing waterbodies since they already exist, and a value of 1 was given to areas without water bodies.

## 3. Results

### 3.1. Analysis of LULC Distribution and Changes in Maseru

In Figure 2, we show the LULC classification across Maseru, Lesotho for the periods 1988, 1993, 1998, 2003, 2008, 2013 and 2019. Our results show a clear landscape pattern, with built-up areas dominating the central northeast section of the study area. From 1988, this spread steadily and, in 2019, consolidated the interior and further south of the city (Figure 2). Against this expansion, the clear decrease in agricultural areas and grassland areas over the years is remarkable. Relatively smaller waterbodies occurred mostly from the central section and northern parts flanked by built-up areas. There were few waterbodies and woody vegetation in the study area (Figure 2). There was also a rise in areas with bare soils from 1993 to 2019 (Figure 2).

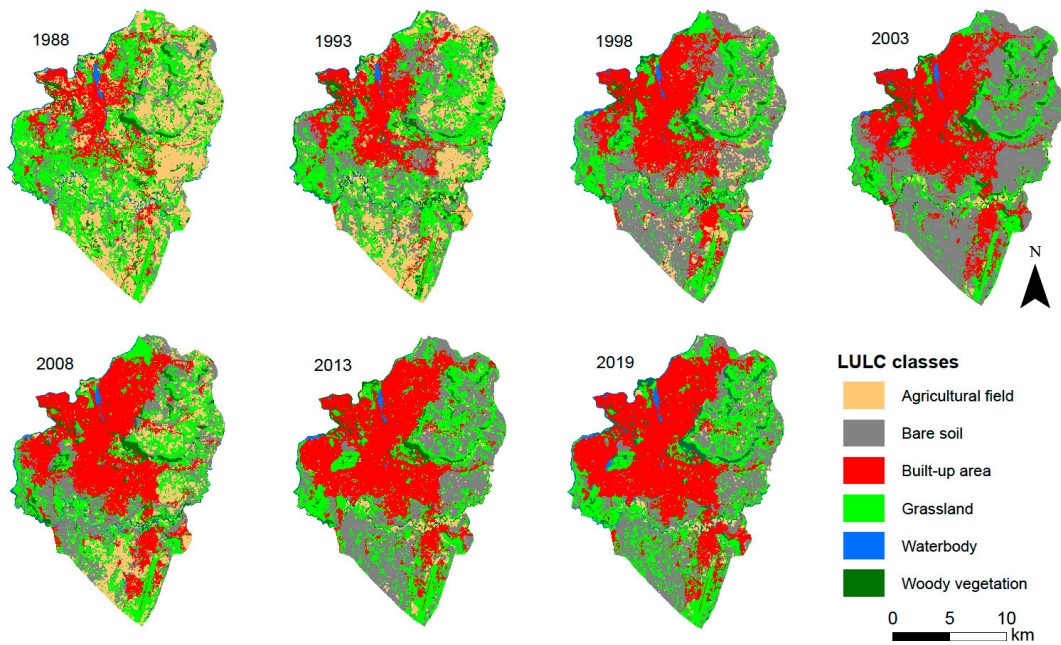

**Figure 2.** LULC classification maps from 1988 to 2019 for the study area.

### 3.2. Quantitative Analysis of LULC Changes in Maseru

Table 3 presents the areal extent for each LULC class in the study area of almost 29,825 ha between 1988 and 2019. Our results show that the landscape of Maseru in 1988 was dominated by two-thirds (63.1%) of natural land cover classes consisting of grassland vegetation (43.3%), woody vegetation (5.5%) water bodies (2%) and bare land (12.3%) and decreased to 10.5%, 3.3%, 1.3%, respectively by 2019, except for bare soil which rose to 35.3%. In 2019, only 15,013 ha remained as natural land cover, suggesting a conversion of 3794 ha to land use activities over the study period. The remaining 11,018 ha of land was occupied by land use activities (built-up area and agricultural land). As of 2019, we noted a tripling of the built-up area from 15.3% in 1988 to 48% in 2019. This is of concern as it is an extremely hard and irreversible type of change, completely obliterating the original natural landscape and unlikely to be reversed to its original natural state. The same period also saw an unexpected decrease in agricultural areas from 21.7% to 1.7% of the city.

**Table 3.** The total area per hectare (H) for different land use and land cover types from 1988 to 2019.

| | | 1988 | | 1993 | | 1998 | | 2003 | | 2008 | | 2013 | | 2019 | |
| | | **Ha** | **%** | **Ha** | **%** | **Ha** | **%** | **Ha** | **%** | **Ha** | **%** | **Ha** | **%** | **Ha** | **%** |
|---|---|---|---|---|---|---|---|---|---|---|---|---|---|---|---|
| LULC classes | AF | 6470 | 21.7 | 6091 | 20.4 | 4901 | 16.4 | 3161 | 10.6 | 1031 | 3.5 | 947 | 3.2 | 510 | 1.7 |
| | BS | 3671 | 12.3 | 6246 | 20.9 | 8347 | 28.0 | 9542 | 32.0 | 9850 | 33.0 | 10,252 | 34.4 | 10,538 | 35.3 |
| | BU | 4548 | 15.3 | 5739 | 19.2 | 8226 | 27.6 | 9265 | 31.1 | 11,515 | 38.6 | 12,520 | 42.0 | 14,302 | 48.0 |
| | GL | 12,911 | 43.3 | 9656 | 32.4 | 6341 | 21.3 | 6003 | 20.1 | 5618 | 18.8 | 4557 | 15.3 | 3138 | 10.5 |
| | WB | 583 | 2.0 | 453 | 1.5 | 424 | 1.4 | 422 | 1.4 | 413 | 1.4 | 401 | 1.3 | 381 | 1.3 |
| | WV | 1642 | 5.5 | 1641 | 5.5 | 1586 | 5.3 | 1432 | 4.8 | 1398 | 4.7 | 1148 | 3.8 | 956 | 3.2 |

Agricultural field (AF), bare soil (BS), built-up (BU), grassland (GL), waterbody (WB), woody vegetation (WV).

### 3.3. Validation of LULC Maps

We validated the classified maps for 1988, 1993, 1998, 2003, 2008, 2013 and 2019 using test samples (30%). The overall accuracies for all the maps ranged from 88% (1998) to 95% (2008), as shown in Table 4. The kappa index values for the LULC maps used in this study ranged from 0.8 (1998) to 0.9 (2008), as shown in Table 4. The user's accuracy for all the years of study ranged from 67% for the agricultural fields in 1998 to 100% for the woody vegetation in 2008 as well as for the waterbody (2013) and bare soil (2019) (Table 4). The producer's accuracies for the LULC maps raged from 40% for the agricultural field class in 1998 to 100% for the bare soil class in 2003 and 2013 (Table 4).

**Table 4.** Confusion matrices for land use and land cover classification for Maseru city (1988–2019).

| Year | | **Classes** | | | | | | OA | KI |
| | | AF | BS | BU | GL | WB | WV | | |
|---|---|---|---|---|---|---|---|---|---|
| | Reference total | 48 | 61 | 76 | 54 | 38 | 32 | | |
| | Classified | 47 | 54 | 90 | 48 | 39 | 31 | | |
| 1988 | Correctly classified | 42 | 52 | 72 | 46 | 35 | 27 | 89 | 0.86 |
| | UA (%) | 88 | 85 | 92 | 85 | 82 | 84 | | |
| | PA (%) | 89 | 96 | 80 | 96 | 90 | 87 | | |
| | Reference total | 43 | 54 | 78 | 47 | 36 | 42 | | |
| | Classified | 42 | 56 | 90 | 46 | 33 | 35 | | |
| 1993 | Correctly classified | 40 | 49 | 75 | 43 | 32 | 33 | 90 | 0.88 |
| | UA (%) | 90 | 91 | 96 | 91 | 89 | 79 | | |
| | PA (%) | 95 | 88 | 83 | 93 | 97 | 94 | | |
| | Reference total | 9 | 52 | 98 | 34 | 33 | 28 | | |
| | Classified | 15 | 51 | 94 | 36 | 33 | 28 | | |
| 1998 | Correctly classified | 6 | 48 | 90 | 31 | 29 | 22 | 88 | 0.84 |
| | UA | 67 | 92 | 92 | 91 | 88 | 79 | | |
| | PA (%) | 40 | 94 | 96 | 86 | 88 | 79 | | |

**Table 4.** *Cont.*

| Year | | AF | BS | BU | Classes GL | WB | WV | OA | KI |
|------|------|----|----|----|----|----|----|----|----|
| 2003 | Reference total | 14 | 58 | 94 | 36 | 26 | 37 | | |
| | Classified | 18 | 57 | 92 | 34 | 23 | 37 | | |
| | Correctly classified | 12 | 57 | 88 | 29 | 20 | 34 | 91 | 0.88 |
| | UA (%) | 86 | 98 | 94 | 81 | 92 | 92 | | |
| | PA (%) | 67 | 100 | 96 | 85 | 87 | 92 | | |
| 2008 | Reference total | 33 | 52 | 96 | 35 | 28 | 30 | | |
| | Classified | 32 | 54 | 92 | 33 | 30 | 33 | | |
| | Correctly classified | 31 | 50 | 90 | 32 | 27 | 30 | 95 | 0.94 |
| | UA (%) | 94 | 96 | 94 | 91 | 96 | 100 | | |
| | PA (%) | 97 | 93 | 98 | 97 | 90 | 91 | | |
| 2013 | Reference total | 22 | 59 | 85 | 42 | 28 | 43 | | |
| | Classified | 33 | 53 | 90 | 37 | 29 | 40 | | |
| | Correctly classified | 20 | 53 | 83 | 31 | 28 | 39 | 92 | 0.90 |
| | UA (%) | 91 | 90 | 98 | 74 | 100 | 91 | | |
| | PA (%) | 61 | 100 | 92 | 84 | 97 | 98 | | |
| 2019 | Reference total | 22 | 52 | 94 | 23 | 38 | 36 | | |
| | Classified | 25 | 53 | 90 | 28 | 37 | 32 | | |
| | Correctly classified | 20 | 52 | 85 | 20 | 36 | 30 | 89 | 0.86 |
| | UA (%) | 91 | 100 | 90 | 87 | 95 | 83 | | |
| | PA (%) | 80 | 98 | 94 | 71 | 97 | 94 | | |

### 3.4. Urban Growth and LULC Change Analysis

The highest rate of growth was observed for bare soil (73.57%) between 2013 and 2019 (Figure 3). There was the highest decrease (−41%) in land covered by agricultural fields between 2013 and 2019 (Figure 3). The built-up area increased over the years from 1988 to 2019, with the highest figure of 40.67% between 2013 and 2019 (Figure 3). A detailed bar graph (Figure 3) shows the gains and losses for each class within the years from 1988 to 1993, 1993 to 1998, 1998 to 2003, 2003 to 2008, 2008 to 2013 and 2013 to 2019.

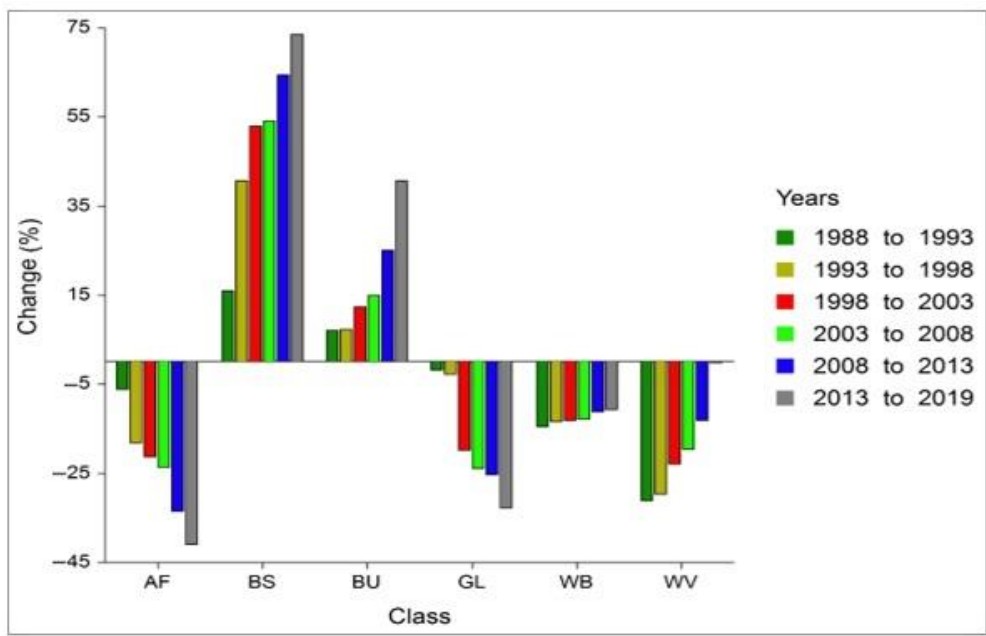

**Figure 3.** Changes in percentage (%) for land use and land cover (LULC) classes in the study area between 1998 to 2019. The classes are agricultural field (AF), bare soil (BS), built-up (BU), grassland (GL), waterbody (WB) and woody vegetation (WV).

### 3.5. Analysis of Spatial Variables

We simulated future urban growth using spatial variables (slope map, roads with a 50 m buffer, water bodies and existing built-up areas in 2019). We assigned the map with a slope percentage of <25%, a value of 1, and 0 represented an unsuitably steep slope (Figure 4a). For the roads and water bodies maps, existing ones were rated 0 (unsuitable) and 1 for areas without roads and water bodies, respectively (Figure 4b,c). The existing built-up areas in 2019 were given a value of 0 (unsuitable), and the undeveloped areas were given a value of 1 (suitable), as shown in Figure 4d.

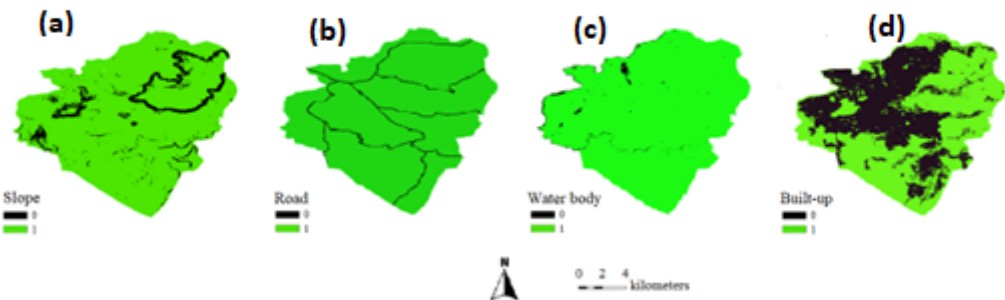

**Figure 4.** Spatial variables used in the prediction of urban growth. The spatial variables are for: (**a**) Slope, (**b**) Road, (**c**) Water body and (**d**) Built-up.

### 3.6. Model Performance Validation

We used the validation module in MOLUSCE to validate the 2019 LULC based on the simulated map derived from the 2008 and 2013 LULC maps. These two years were selected based on the high classification accuracy achieved. The validation model calculated the percentage of correctness, the overall kappa, the kappa histogram and the kappa location. Our results from the validation module are shown in Table 5. They show high values that made the model suitable for predicting LULC changes in the study area.

**Table 5.** Validation of the Predicted model using Kappa. 2019 LULC map was simulated using 2008 and 2013 classified images.

| Parameter | Value (%) |
| :---: | :---: |
| Kappa (histogram) | 88 |
| Kappa (location) | 89 |
| Kappa (overall) | 84 |
| Percentage (%) of correctness | 87 |

### 3.7. Urban Growth Prediction for 2050

We used the spatial variables (Figure 4) and LULC maps for 1988 and 2019 to predict changes in 2050, accounting for the 1988 and 2019 temporal differences of 31 years in the input LULC. Our results show an increase in the built-up area covering more than half of the study area (Figure 5).

The other LULC classes, such as grassland, agricultural land, water body, and woody vegetation, were projected to decrease in their coverage area in 2050 (Table 6). The highest rate of decrease was expected for grassland ($-35.8\%$), followed by bare soil ($-33.6\%$), agricultural field ($-22.3\%$), water body ($-2.2\%$) and woody vegetation ($-0.7\%$) as shown in Table 6. The built-up area was expected to increase by 5434.5 ha in 2050, from 11,973.2 ha in 2019 to 17,407.7 ha in 2050 (Table 6). The smallest change was expected for the water body class, with an expected decrease from 1506.1 ha in 2019 to 1495.9 ha in 2050 (Table 6). The expected LULC coverage changes for the year 2050 are presented in Table 6.

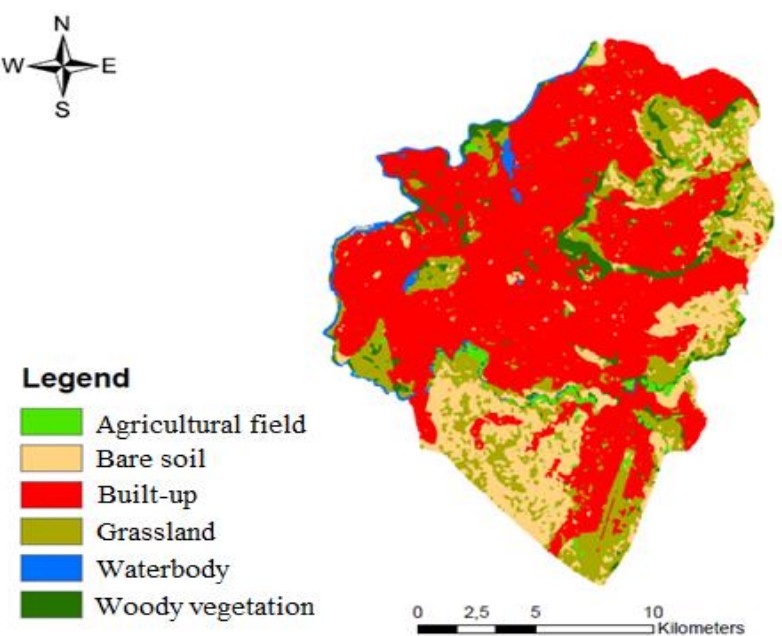

**Figure 5.** Predicted land use and land cover (LULC) map for 2050.

**Table 6.** The predicted changes in The built-up area and other LULC types from 2019 to 2050.

| Class | 2019 | | 2050 | | Predicted Change 2019–2050 | |
|---|---|---|---|---|---|---|
| | **ha** | **%** | **ha** | **%** | **ha** | **%** |
| AG | 847.1 | 2.8 | 657.8 | 2.2 | −189.3 | −22.3 |
| BS | 6939.6 | 23.3 | 4609.8 | 15.5 | −2328.8 | −33.6 |
| BU | 11,973.2 | 40.1 | 17,407.7 | 58.4 | 5434.5 | 45.4 |
| GL | 8139.8 | 27.3 | 5228.8 | 17.5 | −2910.0 | −35.8 |
| WB | 421.3 | 1.4 | 412.4 | 1.4 | −9.1 | −2.2 |
| WV | 1506.1 | 5.1 | 1495.9 | 5.0 | −10.2 | −0.7 |

## 4. Discussion

### 4.1. LULC Maps

This study assessed the impact of urbanization on LULC changes in the study area. Evidence of urbanization is evident in the increasing area of coverage for built-up areas, with the highest growth of 8.2% between 1993 and 1998 (Table 3). This growth was mainly due to the industrialization in Maseru, where the first textile factories were built in the 1990s. The emergence of these textile factories was largely a result of the collapse of Apartheid in South Africa, which freed up some policies that enabled countries such as Lesotho to open in the 1990s [51]. There was a massive movement of people from rural areas to obtain employment in the new textile factories in Maseru [52]. Moreover, the massive retrenchment of migrant mine workers in nearby South Africa in the early 1990s led more people to move from rural areas in Lesotho to Maseru in search of employment and better living conditions [53]. In 1993 and 1998, general elections were conducted in Lesotho, which led to a massive return of political exiles, most of whom built and resided in Maseru: the economic hub of the country [52]. The continuous rise in the built-up areas from 1988 to 2019 was due to population growth, which rose from approximately 311,829 in 1986 to 519,186 in 2016 [54].

Many researchers have found a similar link between high urbanization and economic growth for the provision of housing and basic infrastructure development, especially in Sub-Saharan countries [14,15,55,56]. This inequitable growth has resulted in various problems, which include traffic congestion, road accidents, and air, land and water pollution [20]. The city is also facing extreme poverty, with high rates of unemployment and hunger [57].

### 4.2. Accuracy Assessment for LULC Maps

The SVM classifier performed well in the classification of Landsat imageries with overall accuracies ranging from 88% to 95%, which are above the minimum acceptable value of 85% recommended by Anderson [58]. These accuracies are due to the strengths of the SVM: a non-parametric classifier that is effective in handling any numerical data without assuming data distribution [32]. However, low producer accuracy (40%) and user accuracy (67%) were observed for the agricultural field class in 1998. This was maybe due to the limited number of samples that were used for training (38) and testing (15), as shown in Table 2. The limited number of samples for the agriculture class was due to the small size of agricultural land in the urban area. In the future, it is recommended to have more samples to improve the accuracy of LULC classification. This is in line with Morales-Barquero et al. [59], who stated that having a high number of sampling units is recommended to improve accuracies in remote sensing studies. It is also worth noting that the heterogeneity and topographic nature of Maseru also had an impact on the production of low accuracies for the classified maps. There was the "mixed-pixel problem" where some pixels were not entirely occupied by one class [60], considering that Landsat imageries with a 30 m resolution were used. This resulted in the misclassification of features with lengths and widths less than 30 m. The use of very high spatial resolution imageries such as WorldView-3 and Pléiades could be used to produce high accuracies in future studies.

### 4.3. Interannual Changes in LULC Classes

The highest growth of 73.6% in the bare soil class was observed between 2013 and 2019, whilst the highest decrease (−41%) was for areas covered with agricultural fields. This decline in agricultural fields might be a result of an increase in settlements due to a rise in population and road development over the years. This is in line with Maro [21], who highlighted that increased road development and settlements in 2000, 2001, 2005 and 2007 brought LULC changes in Maseru. This is also supported by the World Bank [20], which highlighted that the rise in population led to the building of houses and the construction of roads, resulting in negative effects such as pollution and undesirable LULC changes.

The decrease in agricultural fields (−6.1%) and an increase in bare soil (15.9%) between 1988 and 1993 (Figure 3) could also be attributed to poor farming methods and the 1992 severe drought which faced all Southern African countries. During this drought period, there was low rainfall and high temperatures, which reduced agrarian produce, where most agricultural fields had no crops, leading to a food security crisis in Southern Africa [61]. Rantšo supports this, and Seboka [62] highlighted that similar to other Southern African countries, Lesotho faced a food crisis and land degradation due to population growth in the country over the years and during extreme drought events such as the severe 1992 and 2015 drought. The population in Maseru was estimated to be around 150,000 in 1989, which rose to approximately 519,186 in 2016 [63].

During the study period, the amount of land covered by built-up structures increased from 15.3% in 1988 to 48% in 2019 (Table 3). This increase in the built-up area could be attributed to the internal migration of people from rural areas to Maseru and from nearby South Africa. People migrated to Maseru for jobs in textile factories, local income-earning opportunities in the informal sector, and a decline in the South African demand for unskilled labor [64]. The rising population in Maseru has resulted in urban expansion and in-filling, where agricultural farmlands have been converted into residential development [53]. This urbanization process also led to a decrease in areas covered by grasslands, water bodies and woody vegetation (Figure 3).

### 4.4. Prediction of Urban Growth

The prediction results for the year 2050 (Figure 5 and Table 6) show that there will be LULC changes in the study area that are mainly due to urbanization. There will be a reduction in the area covered by agricultural fields, bare soil, grassland, water bodies and woody vegetation (Table 6). There will be an increase in the area covered by built-up

structures due to the current continuation of urban dwellers and projected population growth in Maseru. This is supported by Crush [64], who states that Maseru will continue to develop and build new infrastructure to cater to the rising population and migrants from rural communities and South Africa. This projection shows that the population of Maseru is expected to rise from 519,186 in 2016 to about 716,773 in 2036 [63]. This is consistent with the United Nations [2], which noted that urbanization in Lesotho has risen, and the urban population is expected to rise from 39% by 2025 to 58% by 2050. Other LULC classes are expected to decrease to accommodate this tremendous growth of areas covered by built-up structures.

## 5. Conclusions

Our results show that urbanization has been the main driver of LULC changes in the city of Maseru, Lesotho, between 1988 and 2019. Our results have validated the value of Landsat data with spatial variables and the MOLUSCE model, which effectively simulated and predicted the LULC changes in Maseru city. Based on these results, we concluded that:

1. Landsat products provide satisfactory results in classifying and mapping LULC changes from 1988 to 2019. The overall accuracy ranged from 88% to 95%, with kappa values between 0.84 and 0.94.
2. Remarkable LULC changes occurred in Maseru from 1988 to 2019, with the built-up area increasing from 15.3% to almost half (48%) of the city, much of which consumed pristine classes such as agricultural lands and grasslands.
3. In 2050, built-up areas are projected to increase further, while the area covered by agricultural land, bare soil, water bodies and woody vegetation is expected to decrease.

Overall, our study provides useful insights for land management authorities in the city of Maseru, Lesotho, so that proactive planning strategies can be formulated to achieve sustainable development. Our results are also valuable for implementing the Maseru 2050 urban plan. Using high-resolution imagery such as unmanned aerial vehicles (UAV)-drone-derived LULC is recommended to provide more detailed spatial data for urban planning. However, the use of UAVs is limited by the large area, big data processing capacity, and legal regulation of UAV operations. Future studies should consider economic factors in the simulation of LULC in 2050 and beyond and also incorporate the latest deep-learning methods.

**Author Contributions:** Conceptualization, N.E.M. and E.A.; methodology, validation, and formal analysis N.E.M.; writing—original draft preparation, review and editing, N.E.M., E.A. and S.X.; visualization, S.X.; supervision, E.A. All authors have read and agreed to the published version of the manuscript.

**Funding:** This research received no external funding.

**Institutional Review Board Statement:** Not applicable.

**Informed Consent Statement:** Not applicable.

**Data Availability Statement:** Data are available upon request.

**Acknowledgments:** The authors would like to thank the United States Geological Survey (USGS) for processing and providing Landsat data. The authors thank the reviewers for their constructive feedback.

**Conflicts of Interest:** The authors declare no conflict of interest.

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
