# Peer review of "Spatial Assessment and Prediction of Urbanization in Maseru Using Earth Observation Data"

_applsci, doi:10.3390/app13105854_

Round 1

Reviewer 1 Report

Observation Data

Thank you for the opportunity to review the paper “ Spatial Assessment and Prediction of Urbanization in Maseru 2 Using Earth Observation Data”. Overall, I found that the paper is written well.  I have the following few comments

 Comments to the Authors

Line 20 Not a surprising result, its everywhere where new urban areas have been developed at the cost of agricultural/forest land

Line 23 Don't expect the urban government will impost some policies to regulate such expansion of urban areas? This is your assumption  based on ceteris paribus

Line 55 LULC changes along the periphery of the urban areas are not natural; cities have a greater role in regulating it before substantial changes occur due to anthropogenic reasons. Please discuss the city’s past and current attempts to address urbanization and the current ordinance and zoning policies to convert lands into urban areas.

Line 86- I suggest discussing the benefits of ANN-CA over LTM and CLUE. Why you chose ANM-CA? Need to add a few literatures here discussing the benefits of ANN-CA

Line 89- It could be the other way too, LULC impacts urbanization as urbanization is not a natural phenomenon. Please clarify why you think it's a unidirectional impact.

Line 101 Should you say objectives instead of goals or goals?

Line 123-24 How did you address the three months temporal gaps in some data during the study period? There might be variations in December and March in some of the agricultural lands, and their spectral properties confound with bare soil.

Line 142-143 - What were the standard criteria or definitions to distinguish these classes from each other? Did you follow Anderson’s Classification?

Line 146 Would you provide a basis for splitting into this ratio for training and classification of all images?  Are these points stationary (constant) for all images? If not, why did you not use the constant points?

Line 146- Could you also include a map of these samples showing the points selected

Line 241-243  Should you also consider using City or government regulations and ordinances?

Line 286 Could you explain a bit more about the use of reference maps in validating your classified maps? What were the references maps, and how did you ensure their validity to present close dates of image acquisition dates? Did you digitize those aerial maps from the 1980s and 1990s? It is not clear.

Line 303, Why was bare soil increased? Explain

Line 321, Specific reason for using 2008 and 2013 years for model variations?

In 376 Any reason why you were not able to have more samples for accuracy assessment of the agricultural field?

 Line 381- What percentage of your study area had the mixed pixel issue? How did you handle it? It is not clear.

Line 403, was the drought impact lasted until 2019? Since bare soil increased in all periods from the 1980s to 2019 per Fig 3

Line 438 Change to 15.3%

 Line 444-

In conclusion, I would suggest providing examples of policy interventions or regulations to discourage to mitigate the projected increase in urban areas.  I also suggest the use of evolving UAV-drone-derived high-resolution imagery to increase the power of the model. 

Author Response

Reviewer 1

Thank you for the opportunity to review the paper “ Spatial Assessment and Prediction of Urbanization in Maseru 2 Using Earth Observation Data”. Overall, I found that the paper is written well.  I have the following few comments

Thank you for this comment, much appreciated.

Line 20 Not a surprising result, its everywhere where new urban areas have been developed at the cost of agricultural/forest land

We agree, increase of built-up area at the expense of agricultural land, grassland and forest vegetation is a common pattern. We have added statistics to give the overall picture of the changes in Maseru since the degree of changes are unique to specific target areas. The sentence now reads, “Our results show a notable increase in the built-up area from 15.3% in 1988 to 48% in 2019 and bare soil from 12.3% to 35.3%, while decreased agricultural land (21.7 to 1.7%), grassland (43.3 to 10.5%) and forest vegetation (5.5 to 3.2%) was observed over the study period”.

Line 23 Don't expect the urban government will impost some policies to regulate such expansion of urban areas? This is your assumption  based on ceteris paribus

We based our assumption on Maseru 2050 Urban Plan available from https://www.gov.ls/maseru-2050-urban-plan/. For the purposes of this plan, detailed analysis of LULC is required and our study partly provide such information.  

Line 55 LULC changes along the periphery of the urban areas are not natural; cities have a greater role in regulating it before substantial changes occur due to anthropogenic reasons. Please discuss the city’s past and current attempts to address urbanization and the current ordinance and zoning policies to convert lands into urban areas.

the city’s past and current attempts to address urbanization has been discussed in the introduction section as recommended by the reviewer.

Line 86- I suggest discussing the benefits of ANN-CA over LTM and CLUE. Why you chose ANM-CA? Need to add a few literatures here discussing the benefits of ANN-CA

We added literature describing the advantages of ANN-CA, and it reads “It has been found to be effective for analyzing nonlinear complex LULC phenomena and avoid the automatic acquisition of conversion rules during transitional computations [18,19]. Using its self-organization, self-learning, association, and memory, ANN is capable of simplifying the acquisition of CA model conversion rules, extracting CA conversion rules from the original training data, and eliminating subjective factors, thus improving simulation accuracy [18]”.

Line 89- It could be the other way too, LULC impacts urbanization as urbanization is not a natural phenomenon. Please clarify why you think it's a unidirectional impact.

Urbanization have positive as well negative impact on LULC in countries in sub-Saharan Africa, which includes Lesotho [20]

Line 101 Should you say objectives instead of goals or goals?

We changed “goal” to “objective”

Line 123-24 How did you address the three months temporal gaps in some data during the study period? There might be variations in December and March in some of the agricultural lands, and their spectral properties confound with bare soil.

With Landsat temporal acquisitions, particularly in tropical regions, it is difficult to obtain cloudless images falling in the same month. We therefore extended the period to cover the rainy season from December to March, with the aim to distinguish between bare areas and vegetated and built-up areas.

Line 142-143 - What were the standard criteria or definitions to distinguish these classes from each other? Did you follow Anderson’s Classification?

Classes were defined from the Lesotho Land Cover Atlas developed by FAO, and it now reads “The LULC classes were defined from the Lesotho Land Cover Atlas developed by the Food and Agriculture Organization (FAO) of the United Nations [30]”.

Line 146 Would you provide a basis for splitting into this ratio for training and classification of all images?  Are these points stationary (constant) for all images? If not, why did you not use the constant points?

The splitting ratio (70%/30%) is commonly used in remote sensing image classification.   We have added the following to the sentence “To test the performance of the classification model,  the sample points for the multitemporal images were divided into 70% (training) and 30% (test) for the classification of all the imageries”

Line 146- Could you also include a map of these samples showing the points selected

We thank the reviewer for this valid suggestion. However, as you can see from Table 2, different numbers of samples were generated for each year. We will need seven other maps to cover the study period. Therefore, we prefer to explain the sampling strategy using a table format instead.

Line 241-243  Should you also consider using City or government regulations and ordinances?

Great point! We only focused here on the physical factors that drive the urban expansion.

Line 286 Could you explain a bit more about the use of reference maps in validating your classified maps? What were the references maps, and how did you ensure their validity to present close dates of image acquisition dates? Did you digitize those aerial maps from the 1980s and 1990s? It is not clear.

The reference data  (training and the test ) was generated from google earth images using visual interpretation. The google earth images were selected from the archive to correspond with Landsat image acquisition dates. The predicted maps were tested against the test dataset to validate the classification results, and a confusion matrix was generated to calculate the accuracy assessment.

Line 303, Why was bare soil increased? Explain

This is due to a decrease on other classes such grassland, woody vegetation and waterbody due to drought and rainfall variability.

Line 321, Specific reason for using 2008 and 2013 years for model variations?

We have added “these two years were selected based on the high classification accuracy achieved”

In 376 Any reason why you were not able to have more samples for accuracy assessment of the agricultural field?

Due to the small agriculture areas. This has been explained in the discussion section.

Line 403, was the drought impact lasted until 2019? Since bare soil increased in all periods from the 1980s to 2019 per Fig 3

Rantšo supports this, and Seboka [57] highlighted that like the rest of Southern African countries, Lesotho faced food crisis and land degradation due to population growth in the country over the years, and during extreme drought events such as since the severe 1992 drought

Line 438 Change to 15.3%

Thank you, we changed 15,3% to 15.3%

 Line 444-

In conclusion, I would suggest providing examples of policy interventions or regulations to discourage to mitigate the projected increase in urban areas.  I also suggest the use of evolving UAV-drone-derived high-resolution imagery to increase the power of the model.

We rephrased the statement to be more specific. It now reads, “Our results are also valuable for implementing Maseru 2050 urban plan”. Using high-resolution imagery such as unmanned aerial vehicles (UAV)-drone-derived LULC is recommended to provide more detailed spatial data for urban planning. However, the use of UAVs is limited by the large area, big data processing capacity, and legal and regulation of UAV operation.

Reviewer 2 Report

It becomes necessary to improve the quality of the images inserted in the document.

Author Response

Reviewer 2

It becomes necessary to improve the quality of the images inserted in the document.

Thank you for your comment, we have improved the quality of the images

Reviewer 3 Report

1. The description “To our knowledge, this is the first study to provide essential spatial data in urban growth to planners and other stakeholders in Maseru” in the abstract is inappropriate. There has been a large amount of research on global LULC change monitoring, such as GlobalLand30 (China), and the Global Land Survey (GLS), a collaboration between the University of Maryland and the United States Geological Survey (USGS).

2. The significance of the research results in the abstract is not reflected.

3. The area of Maseru is small. Why is Landsat image used in this paper instead of Sentinel 2 satellite image with higher accuracy or other remote sensing images with high resolution?

4. The remote sensing data in Part2.2 is obtained from Landsat remote sensing images of Maseru during the rainy season. The rainy season has a great influence on the quality of remote sensing images, and the remote sensing images of the clear and cloudless period are generally chosen to invert the land cover LULC, why remote sensing images of the rainy season are chosen in this paper.

5. Land cover sample library needs to be shown as a picture in the text.

6. The role of economy on urban growth is significant, and the paper mentions that Maseru's urban expansion is due to industrialization, but economic factors are not considered in the simulation of land cover in 2050, so the credibility of the simulation results in 2050 is doubtful.

7. The land cover classification method is SVM, which is a more traditional method and a supervised classification method that comes with the ENVI software, so why not use deep learning methods like the 2050 land cover simulation?

8. The discussion section goes further into the implications of the 2050 land cover simulation results.

9. For the time span why only until 2019 and not the latest 2022?

10. All the figures require higher image resolution.

11. Inconsistent labeling of references, such as Line 167-178.

12. Formula 4 is not standardized.

Author Response

  1. The description “To our knowledge, this is the first study to provide essential spatial data in urban growth to planners and other stakeholders in Maseru” in the abstract is inappropriate. There has been a large amount of research on global LULC change monitoring, such as GlobalLand30 (China), and the Global Land Survey (GLS), a collaboration between the University of Maryland and the United States Geological Survey (USGS).

Yes, there are existing studies, but not a detailed that quantify and describe long term changes like we did. We therefore rephrased the statement to emphasize this point. It now reads “To our knowledge, this is the first detailed, long-term study to provide insights on urban growth to planners and other stakeholders in Maseru in order to improve the implementation of Maseru 2050 urban plan”.

  1. The significance of the research results in the abstract is not reflected.

Linked to the above comment, we rephrased the sentence “To our knowledge, this is the first detailed, long-term study to provide insights on urban growth to planners and other stakeholders in Maseru in order to improve the implementation of Maseru 2050 urban plan” to also highlight the significance of the results in the abstract.

  1. The area of Maseru is small. Why is Landsat image used in this paper instead of Sentinel 2 satellite image with higher accuracy or other remote sensing images with high resolution?

Yes, we agree, Sentinel-2 would be much better, but the only problem is short temporal coverage that made us be select Landsat to go as far as 1989

  1. The remote sensing data in Part2.2 is obtained from Landsat remote sensing images of Maseru during the rainy season. The rainy season has a great influence on the quality of remote sensing images, and the remote sensing images of the clear and cloudless period are generally chosen to invert the land cover LULC, why remote sensing images of the rainy season are chosen in this paper.

With Landsat temporal acquisitions, particularly in tropical regions, it is difficult to obtain cloudless images falling in the same month. We therefore extended the period to cover the rainy season from December to March, with the aim to distinguish between bare areas and vegetated and built-up areas.

  1. Land cover sample library needs to be shown as a picture in the text.

We thank the reviewer for this suggestion. We believe that the spectral library for landcover types studied here is very well established and won't add much information to the reader.

  1. The role of economy on urban growth is significant, and the paper mentions that Maseru's urban expansion is due to industrialization, but economic factors are not considered in the simulation of land cover in 2050, so the credibility of the simulation results in 2050 is doubtful.

We added a section spelling out the Limitations of the study before conclusions. We highlighted the limitations including the exclusion of economic factors in the simulation of land cover in 2050.

  1. The land cover classification method is SVM, which is a more traditional method and a supervised classification method that comes with the ENVI software, so why not use deep learning methods like the 2050 land cover simulation?

The objectives of the paper were on the use of SVM LULC classification and the ANN-CA for 2050 land cover simulation. We planning to incorporate deep learning methods in subsequent papers.

  1. For the time span why only until 2019 and not the latest 2022?

Thanks for this valid point! The study was conduct in 2020.

  1. All the figures require higher image resolution.

Thank you, the figures will be improved

  1. Inconsistent labeling of references, such as Line 167-178.

Thank, we corrected the reference

  1. Formula 4 is not standardized.

We inserted end bracket

Round 2

Reviewer 1 Report

Thank you for providing me with a chance to review the revised manuscript.  The authors have revised the manuscript addressing my comments. The revision is satisfactory.  The only concern I have is combining the "limitation" either with the discussion section or with the conclusion. I don't suggest having it as a standalone subheading. 

Author Response

Reviewer 1

The only concern I have is combining the "limitation" either with the discussion section or with the conclusion. I don't suggest having it as a standalone subheading. 

The limitation has been combined with the discussion as recommended by both reviewers

Reviewer 3 Report

1. The quality of the pictures needs to be improved and the legends are not clear, such as Fig1, Fig2.

2. Poor legibility of Fig.3, please look at the literature on land use change mapping.

3. Land cover sample library needs to be listed in the Supplementary Material.

4. “5.Limitations” section is not specific.

Author Response

Reviewer 3

  1. The quality of the pictures needs to be improved and the legends are not clear, such as Fig1, Fig2.

The quality has been improved as recommended by the reviewer.

  1. Poor legibility of Fig.3, please look at the literature on land use change mapping.

The figure has been improved.

  1. .Land cover sample library needs to be listed in the Supplementary Material.

As indicated in the manuscript, all the material and data used will be available upon request.

  1. Limitations” section is not specific.

The limitation has been combined with the discussion as recommended by both reviewers.